# Assessment of a Novel Fixation Method of a Frameless Intrauterine Contraceptive Device Inserted during Cesarean Delivery as a Means of Preventing Displacements and Expulsions: A Prospective Observational Study

**DOI:** 10.3390/life12010083

**Published:** 2022-01-07

**Authors:** Hazal Kutlucan, Recep Onur Karabacak, Stefanie De Buyser, Ahmet Erdem, Nuray Bozkurt, Erhan Demirdağ, Dirk Wildemeersch

**Affiliations:** 1Department of Gynecology and Obstetrics, School of Medicine, Gazi University, Ankara 06500, Turkey; okarabacak@gmail.com (R.O.K.); aerdem@gazi.edu.tr (A.E.); nmbozkurt@yahoo.com (N.B.); erhan55_gs@hotmail.com (E.D.); 2Biostatistics Unit, Faculty of Medicine and Health Sciences, Ghent University, 9000 Ghent, Belgium; statcel@ugent.be; 3Contrel Research ICC Technologiepark 82, 9052 Ghent, Belgium; rina.wildemeersch@contrel.be

**Keywords:** fixation, cesarean section, frameless intrauterine device, anchoring, immediate post-placental insertion, postpartum contraception

## Abstract

The primary objective of this study was to assess the novel fixation method of a frameless copper-releasing intrauterine device inserted following placental delivery during cesarean section and analyze its impact in reducing device displacement and expulsion during and after uterine involution. We hypothesized that the dual-anchoring technique could reduce the risk of intrauterine device displacement and expulsion during and after the uterine involution. The study was conducted at the Gazi University Medicine Faculty Hospital in Ankara, Turkey. Twenty-one pregnant women were enrolled. Insertion was performed following placental removal. To confirm the proper placement and good retention of the device, the distance between the fundal serosa (S) and device anchor knot (A) was measured (S–A) during follow-ups, by ultrasound. There were significant differences in the S–A, as observed by ultrasound at discharge and at 6 weeks post-delivery, which is consistent with the tissue contractions associated with uterine involution. Notwithstanding the uterine involution, no device displacements or expulsions occurred, which indicated a good retention of the frameless device. This innovative retention method of the frameless intrauterine device ensures a well-tolerated, long-term contraception, allowing for immediate contraception and proper pregnancy spacing for cesarean scar healing, and overcomes the issue of expulsion encountered with conventional intrauterine systems.

## 1. Introduction

Immediate postpartum contraception plays a pivotal role in ensuring the health, human rights, and wellbeing of women and their babies. Pregnancies within 12 months following childbirth is a known health risk for women and their infants, and the World Health Organization recommends 18–24 months between pregnancies [1]. Recurrent pregnancies <18 months apart following cesarean deliveries increase the risk of uterine rupture [2]. In addition, inadequate spacing between pregnancies has been shown to contribute to increased infant mortality [3,4,5]. Intrauterine devices (IUDs) and hormonal implants are highly effective forms of long-acting reversible contraception (LARC) used to prevent unintended pregnancies [6]. Contraceptive failure rates of IUDs are less than 1%, and IUDs rival permanent tubal sterilization [7]. A study conducted in the UK showed that 32% of women who expressed a desire for sterilization would prefer a reversible contraceptive method instead of irreversible sterilization [8]. However, their use in the postpartum environment is limited by their higher expulsion rates than those of conventional interval insertion. The main drawback of currently available framed IUDs following their postpartum insertion is their high expulsion and displacement rates [9] after either vaginal [10] or cesarean delivery [11]. A recent Cochrane systemic review that included 15 clinical trials revealed that immediate postpartum insertion of T-shaped IUDs overall is safe and convenient, but the expulsion rate is slightly higher compared to delayed insertions or conventional interval insertions [12]. Researchers have attempted to reduce postpartum related expulsions and/or displacements by limiting insertion to 10 min following placental delivery, unfortunately with limited success. Furthermore, early attempts to solve the issue of expulsion by modifying existing devices were not found to have the expected results as, for instance, the addition of two strands of catgut to the horizontal arms of the Delta-TCu 380 framed IUD and a double knotted catgut added in the top of the Gyne-T 380 vertical arm sutured in the myometrium [13]. Displacement of framed devices that rely on size for retention is also associated with device embedment or malposition following postpartum insertion, likely because of uterine involution and associated physiological changes post-delivery [14]. It has been established that the uterine cavity of most women is much smaller than the arm width of T-shaped devices (28–32 mm). A recent study in 410 nulliparous women using 2D or 3D ultrasound demonstrated that the average maximal uterine width is only 22 mm, with 32% of the population having widths below 20 mm [15]. The unwanted effects of postpartum conventional IUD insertions highlight the necessity for further research into the development of different devices or techniques [16]. The issues related to uterine size incompatibility led to the development of frameless devices, which rely on an alternative retention methodology that eliminates the problematic cross-arm with high patient acceptance. Frameless devices are small, flexible, and highly tolerated, allowing for easy uterine adaptation and acceptance in different uterine cavity shapes. The retention mechanism of the frameless devices has been modified for the purpose of immediate post-placental insertion (IPPI) following cesarean section. In 2004, the first version of the frameless intrauterine copper device for insertion during cesarean section was designed. It contained a cone-shaped biodegradable body to help for the device retention in the myometrium. Retention outcomes were advantageous when compared with non-anchored devices; although early removal of this device was difficult, impairing its further use and addressing the need for an improved design and optimized device [17]. The novel retention mechanism of a frameless copper-releasing device was designed to satisfy this need. This fixation technique relies on two anchoring mechanisms. The first is the device anchor knot (A) fixed in the uterine myometrium, and the second is the reassuring suturing of the anchor knot through a biodegradable suture in the uterine serosa (S). The technique may also serve to eliminate risk of displacement and expulsion during and after the uterine involution. The main objective of this study was to assess the novel method of the frameless copper-releasing device inserted during cesarean section with respect to retention and prevention of displacements and expulsions.

## 2. Materials and Methods

### 2.1. Study Design

This study was designed as a prospective, open-label, noncomparative, single-center (Gazi University Medicine Faculty Hospital in Ankara, Turkey) study, which was observational over 3 months with a total of 21 participants. No sample size calculation was performed as this was a pilot, learning-curve study. Ethical approval was granted by the Turkish Ministry of Health Medicines and Medical Devices Agency, in line with the Helsinki Declaration (approval number: 2018-056).

Our hypothesis (H1) is that good retention can be achieved by the novel fixation method of the copper frameless IUD inserted during cesarean section during and after uterine involution. The variable dependent factor was the frameless device retention during and after uterine involution, while the variable independent factor was the novel frameless device fixation method. To standardize the study, all insertions and follow-up evaluations were performed by one investigator (H.K.).

### 2.2. Participant Enrollment

Participants were informed about the contraception method in the third trimester prenatal controls. The inclusion criteria comprised generally healthy, pregnant women scheduled for cesarean section who desired immediate postpartum contraception initiated in during the procedure with an intact and anatomically normal uterus, were willing to remain in the study for 3 months, were able to return for follow-ups, and were mutually monogamous. Women with uterine infection, clinical cervicitis or vaginitis, suspicion of endometrial or uterine pathology, such as congenital malformation of the uterus, large uterine fibromyoma (>3 cm diameter), presence or history of endometrial or cervical malignancy, undiagnosed genital tract bleeding, thromboembolic disorders, stroke, copper allergy diagnosed as Wilson disease, any cardiac, renal, and/or hepatic diseases were excluded. The trial, follow-up visits, benefits and potential risks, the alternative methods of fertility control, and the fact that the participant may withdraw from the study at any time without prejudice were explained for consent. The devices were provided to the participants free of charge.

### 2.3. The Device

Gyn-CS^®^ (Contrel Europe nv, Ghent, Belgium) is an innovative intrauterine concept that combines the method of action and efficacy of other anchored frameless IUDs, with a novel fixation technique. The inserter was designed for transpiercing the fundus and to compensate for postpartum uterine fundus thickness encountered immediately post-delivery and prior to uterine involution that typically occurs at 6 to 8 weeks following delivery [13]. The device is intended for insertion through the uterine incision prior to closure. The inserted part has a novel broad tip allowing for external palpation during the procedure to assure proper placement while preventing inadvertent penetration of the uterus by the inserter tube itself.

The frameless device has no plastic body, making it completely flexible, and it comprises five copper cylinders (quantitative composition: 350 mg copper), each 5 mm in length and 2.2 mm in diameter, threaded on a length of nonabsorbable polypropylene suture material. The product has a proven 5-year duration of effectiveness. The use of copper cylinders allows for almost 100% copper-release area from both internal and external surfaces, allowing for high and long-term effectiveness despite their small size [18]. A tiny preformed surgical knot at the upper end of the device serves as an anchor, keeping the device in place once inserted. The anchor knot includes a hole to pass an external Vicryl^®^ suture and allows for device fixation to the myometrium. Below the anchoring knot, a small stainless-steel tube (2 mm in length and 0.5 mm in diameter) provides clear hyperechogenic visualization of proper device placement by ultrasound (Figure 1).

### 2.4. Insertion Technique

Following placenta removal, the uterus was lifted out of the abdominal cavity. After bleeding control and manually cleaning the inside of the uterus, the applicator was inserted through the uterine incision up to the fundus midline. After the proper fundal position was achieved, the inserter stylet was moved forward, transpiercing the serosa until the anchoring knot placed at the proximal end of device and the stainless-steel tube below it became visible (Figure 2).

On the stainless-steel tube, a clamp was placed to prevent inadvertent retraction. Once clamped, the inserter stylet, followed by the inserter tube, was removed. The noose of the anchor was then threaded with a biodegradable suture material, such as 3–0 suture (Ethicon, Somerville, NJ, USA) or a generic equivalent. The threaded anchor was then retracted below the serosa by exerting traction on the tail of the device through the uterine incision. The passage of the anchor through the denser serosa layers into the myometrium was clearly felt as passing resistance and could subsequently be palpated on the uterine serosa. One end of the Vicryl^®^ absorbable suture was then secured to the serosa and knotted with its other end. Three knots (one normal and two firm) were made directly on the serosa. The anchor knot of the device and three knots made by the inserter constituted the dual anchoring technique (Figure 3).

Finally, the tail was cut at the level of cervix (Appendix A). In some cases, the tail can be preferentially looped into the endometrial cavity [19]. After insertion, the uterine incision was closed.

### 2.5. Follow-Up Examinations

Following insertion, participants were examined at discharge, 6 weeks, and 3 months. Initial gynecological examinations, including cervical visualization, were performed with the help of a two-dimensional (2D) transvaginal ultrasound (Voluson E6) to assess device placement accuracy, and the distance between the uterine serosa and the device anchor (S–A) was measured by means of a stainless-steel tube within the myometrium, under the serosa not in the endometrium, and five copper cylinders in the endometrial hyperechogenicity. The proper appearance provided exclusion of transfundal migration (Figure 4).

### 2.6. Data Collection, Monitoring, and Analysis

Statistical analysis was performed within the Biostatistics Unit of the Faculty of Medicine and Health Sciences at Ghent University. All data were registered to case report forms and excel files. Nominal and continuous data were described using the absolute and relative frequencies and the median and range (minimum–maximum), respectively. The Wilcoxon signed rank test was conducted at the two-sided 5% significance level. Analyses were performed using IBM SPSS Statistics 26. The Wilson Score 95% confidence interval was calculated around the prevalence estimates at the third follow-up of field insertion and perforation, continuation, and expulsion using the “PropCIs” package in R version 6.3.1 (R Core Team, Foundation for Statistical Computing, Vienna, Austria, 2019).

## 3. Results

Twenty-one women were enrolled between March 2018 and July 2019. There are approximately 1300 cesarean sections performed annually in our clinics. All participants were married, of Turkish ethnicity, and 30 years median age (20–38), and 62% had at least high-school education. Other demographics are shown in Table 1.

From the outset, it was planned to follow patients at discharge, 6 weeks, and 3 months. However, this strategy was hampered as three participants (14.28%) only attended the last control visit at 8, 9, and 14 months, respectively (Table 2).

The total sum of follow-up months was 99 (The total database is in Appendix A).

All insertions were successful with no complications. S–A distance was found to decrease significantly between discharge and 6 weeks (*p*-value from Wilcoxon signed rank test in 10 patients = 0.009), while no significant difference was observed between 6 weeks and the exit visit (≥3 months; *p*-value from Wilcoxon signed rank test in 10 patients = 0.62) (Figure 5).

The sutured device from the anchor was properly retained during the full process of uterus involution, and no migration or expulsion occurred. Participants’ continuation rate up to the exit visit was high, with no patients lost to follow-up. No bleeding abnormalities were observed, and all devices were well tolerated. There was no significant change in postpartum hemorrhage, and continuance of lochia and uterus healing was normal for puerperium. No adverse effects such as infection, pain, or perforation occurred. No removal requests or pregnancies were reported during the study period.

## 4. Discussion

LARC represents a significant family planning opportunity, especially if available for use in the postpartum timeframe. Unfortunately, for many women, this may be the only occasion for contact with healthcare providers, especially in low-income and underdeveloped countries. A study found that almost half of the women stated that they had unprotected sexual intercourse prior to their first six-week postpartum visit [20,21]. Another study revealed that 40% of women failed to return for their scheduled postpartum control visit [22]. Our study illustrated the difficulties that some women have with access to routine/scheduled medical care, with 43% of our subjects missing their 3-month visit. Many of these issues can be overcome with LARC, which is specifically designed and approved to address the immediate postpartum period [23].

The number of cesarean sections performed worldwide is rapidly increasing, and the need for contraception with high efficacy, low adverse effects, and long-term usage is directing patients and clinicians toward intraoperative methods [24]. In Turkey, over 50% of deliveries are cesarean [25]. Optimizing the timeframe between cesarean delivery and future deliveries may not only reduce the number of unintended pregnancies but may also facilitate the possibility of vaginal birth after cesarean.

Historically, the displacement and the embedment of most T-shaped devices generally occur during or after uterine involution [14]. In a large-scale study (*n* = 140), with the same follow-up period as this study, 1.4% of expulsions for frameless devices and 11.4% for conventional IUDs were reported [19], while another study showed 3.6% expulsion of frameless devices as compared to 22.2% with T-shaped IUDs [26], and another study declared 1% expulsion of frameless devices [27]. In our trial, proper placement was achieved in all cases, without migration, displacement, or removal request. The dual anchoring technique reduced the risk of displacement and expulsion during and after uterine involution. We believe that attention to the placement technique by trained obstetricians was important, as well as cleaning the inside of the uterus and checking the uterine cavity for uterine abnormalities such as myomas, feeling the passing suture after pushing the device into the myometrium and not the endometrium, and arranging the midline of the uterine fundus for transpiercing, representing contributing factors for optimal results.

This study showed that the device was well tolerated in all women and did not cause abnormal pain, infection, bleeding irregularities, or discomfort. Insertion was found to be uncomplicated and rapid once familiarity with the procedure was gained; the time required for the first six insertions was 6–8 min, which decreased to 3–4 min thereafter.

### Study Strengths and Weaknesses

The findings of our study are consistent with earlier reports confirming the simplicity of the insertion procedure. The procedure was very well tolerated, supporting the concept of postpartum contraception and, thus, allowing women additional contraception options and possible alternatives to surgical sterilization. The weaknesses were the limited number of study participants (*n* = 21), difficulties in patient compliance with follow-up monitoring, and the short follow-up period. Of the patients, 57% attended the 3-month follow-up; however, although delayed, most women returned for their exit visit. These difficulties highlight the importance of postpartum LARC. A direct comparison of expulsion rates between T-shaped and frameless devices was not possible given that only one device type was tested.

## 5. Conclusions

The results confirm that the dual anchoring technique allowed for effective retention, and the device was highly tolerated, allowing for immediate contraception during cesarean delivery, with the assurance that the procedure is fully reversible. This innovative method, when properly executed, showed no risk of expulsion or displacement while providing convenient immediate contraception after cesarean delivery. A uterine device, designed specifically for long-term contraception and inserted during cesarean delivery, will benefit women and their future pregnancies, as the device can minimize unintended pregnancies while also allowing for full uterine healing prior to the next pregnancy. Given the availability of this innovative device and insertion technique for exclusive postpartum use, clinicians will likely recognize the importance of intra-cesarean contraception that offers convenience and access to LARCs or a reversible alternative to permanent sterilization.

## Figures and Tables

**Figure 1 life-12-00083-f001:**
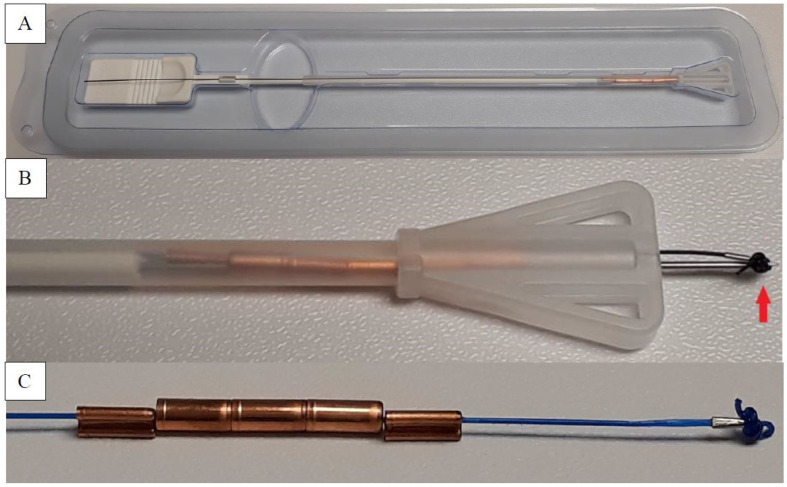
(**A**) The device in blister. (**B**) The inserter tube with triangular part, the anchor knot (arrow), and the stainless-steel tube (under the knot) affixed onto the suture thread. (**C**) The magnified copper-bearing device.

**Figure 2 life-12-00083-f002:**
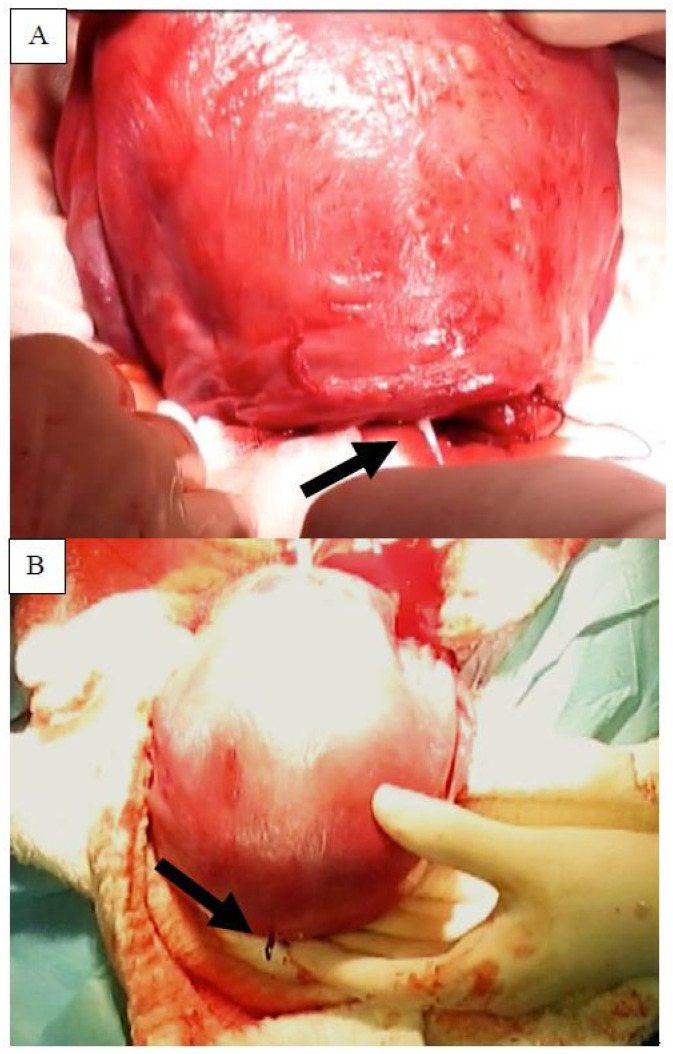
(**A**) Start of insertion from uterine incision after removal of placenta. (**B**) Fundal serosa with transpierced inserter stylet and the knot.

**Figure 3 life-12-00083-f003:**
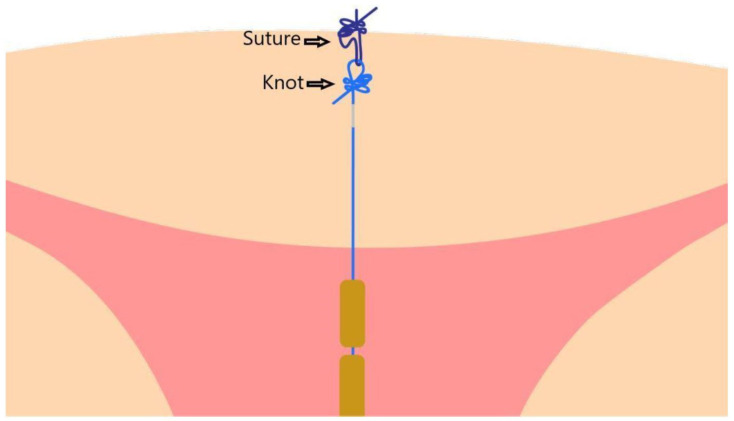
Dual anchoring technique.

**Figure 4 life-12-00083-f004:**
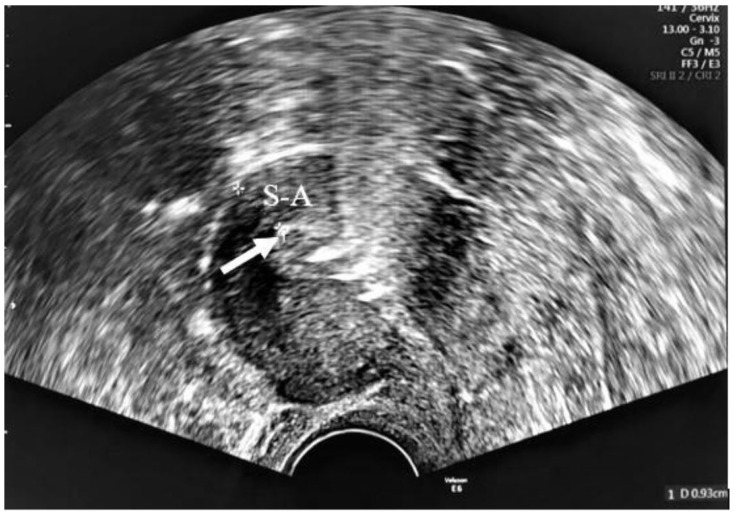
The stainless-steel tube below the anchoring point in the myometrium (arrow) and the separated copper beads are seen as hyperechogenic. Serosa–anchor distance is shown with S–A.

**Figure 5 life-12-00083-f005:**
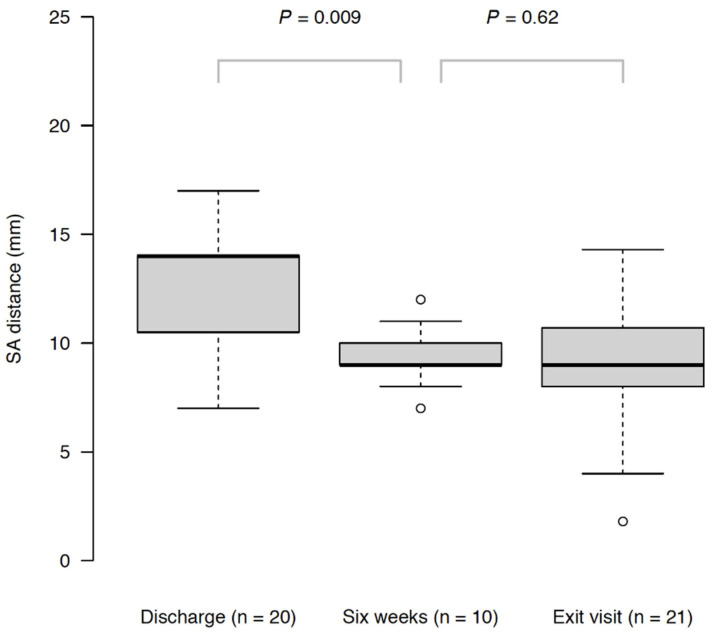
Reported *p* values correspond to two-sided Wilcoxon signed rank tests with continuity correction.

**Table 1 life-12-00083-t001:** Participants’ obstetrical history.

		Count	Column Valid N%
Gravidity/Parity	1-1	2	9.5%
	2-2	6	28.6%
	3-2	4	19.0%
	3-3	5	23.8%
	4-2	2	9.5%
	4-3	1	4.8%
	5-4	1	4.8%
Number of previous CS	0	5	23.8%
	1	12	57.1%
	2	4	19.0%
Reason for CS	Previous caserean section (CS)	16	76.2%
	Maternal disease (history of VSD operation)	1	4.8%
	Maternal disease (history of rectocele operation)	1	4.8%
	Maternal request	3	14.3%

**Table 2 life-12-00083-t002:** Absolute and relative frequency of participants according to events.

Event		Number of Participants	
Insertion		21	
Discharge		20	
6 weeks		10	
Exit visit (months)	3	12 (57.12%)	21
	4	1 (4.76%)	
	5	2 (9.52%)	
	6	3 (14.28%)	
	8	1 (4.76%)	
	9	1 (4.76%)	
	14	1 (4.76%)	

## Data Availability

The data that support the findings of this study are available from the corresponding author, [HK], upon reasonable request.

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
