# Peer review of "Assessment of a Novel Fixation Method of a Frameless Intrauterine Contraceptive Device Inserted during Cesarean Delivery as a Means of Preventing Displacements and Expulsions: A Prospective Observational Study"

_life, 2022, doi:10.3390/life12010083_

Round 1
Reviewer 1 Report
This is a well-written manuscript concerning the application of a novel fixation mode of a frameless contraceptive IUD during CS so as to offer immediate contraception overcoming the issue of expulsion encountered with conventional IUDs.
The study results support the safety and efficacy of the method as there were no complications and the follow-up scans showed proper position of the IUD. The introduction provides sufficient background and includes all relevant references. Well illustrated pictures useful for clinical practice are also included. The results are clearly presented and the adequate statistical analysis supports the conclusions. The text has appropriate length and no major spelling issues could be recognized. There is a limitation concerning the number of women included, but it is expected given the prospective design of the study which strengthens the results anyway. Based on the direct clinical implications of this study and the quality of the paper I believe it fulfills the criteria of Life journal and I support its publication in its current form.
Author Response
I appreciated about your comments and thank you for your reviewing.
Point 1: This is a well-written manuscript concerning the application of a novel fixation mode of a frameless contraceptive IUD during CS so as to offer immediate contraception overcoming the issue of expulsion encountered with conventional IUDs.
Response 1: I thank you for your nice evaluation. We all authors took care about immediate contraception and this noval IUD and technique made us more enthusiastic to work on this field. I am so pleased that your appreciated opinion is that it is a well-written manuscript.
Point 2: The study results support the safety and efficacy of the method as there were no complications and the follow-up scans showed proper position of the IUD. The introduction provides sufficient background and includes all relevant references. Well illustrated pictures useful for clinical practice are also included. The results are clearly presented and the adequate statistical analysis supports the conclusions. The text has appropriate length and no major spelling issues could be recognized. There is a limitation concerning the number of women included, but it is expected given the prospective design of the study which strengthens the results anyway. Based on the direct clinical implications of this study and the quality of the paper I believe it fulfills the criteria of Life journal and I support its publication in its current form.
Response 2: I thank you about your all detailed assessments. The weakness of study is study population as we indicated in discussion and we hope that following the positive results on preventing displacements and expulsions in the study participants , we will get more good results in more women.
Reviewer 2 Report
Dear Authors: excellent work, clear, concise and complete. Congratulations
The paper investigates accurately one of the controversial uses of intrauterine devices. Most experiences are not controlled or studied accurately. Methods are explained in detail as the controls. Although many people discuss this indication an unwanted pregnancy is a real problem and maybe it is an excellent opportunity to prevent it. Most publications are not controlled or not to have a scientific qualification
Paper is plenty of data, that validates the results, but made difficult the read, although it is a style, using an appendix for the data may allow a friendly reading.
Technique solves the main problems associated with the use in pregnant, like malposition of expulsion. Technical details are clear and reproducible for an average obgyn. Pictures are good, although a short video would be better to spread the proper technique. Statistics could introduce some bias by the reduced number of patients, but they are enough to validate the data or to reproduce it in a large sample.
Best regards,
The reviewer
Author Response
I appreciated about your comments and thank you for your reviewing.
Point 1: The paper investigates accurately one of the controversial uses of intrauterine devices. Most experiences are not controlled or studied accurately. Methods are explained in detail as the controls. Although many people discuss this indication an unwanted pregnancy is a real problem and maybe it is an excellent opportunity to prevent it. Most publications are not controlled or not to have a scientific qualification
Response 1: I thank you for your evaluations. Our main aim was to assess the noval intrauterine devices for unwanted pregnancies as immediate contraception during caesarean section.
Point 2: Paper is plenty of data, that validates the results, but made difficult the read, although it is a style, using an appendix for the data may allow a friendly reading.
Response 2: I thank you for your comments and I will attach the database table but the table includes more data that is why that we prefer to create the brief tables.
Point 3: Technique solves the main problems associated with the use in pregnant, like malposition of expulsion. Technical details are clear and reproducible for an average obgyn. Pictures are good, although a short video would be better to spread the proper technique. Statistics could introduce some bias by the reduced number of patients, but they are enough to validate the data or to reproduce it in a large sample.
Response 3: I thank you for your all detailed comments. I will attach a short video on your request.
Reviewer 3 Report
Despite, the Intra Caesarean Fixation of Frameless Copper IUD has been previosly published.
The dual-anchoring technique, is a novel method of IUD fixation, which could be reduced the risk of displacement and expulsion after the uterine involution.
The discussion or comparison with other manuscripts about frameless IUD Gyn-CS® may be an important complementation.
Author Response
I thank you for your comments and recommendations.
Point 1: Despite, the Intra Caesarean Fixation of Frameless Copper IUD has been previosly published. The dual-anchoring technique, is a novel method of IUD fixation, which could be reduced the risk of displacement and expulsion after the uterine involution. The discussion or comparison with other manuscripts about frameless IUD Gyn-CS® may be an important complementation.
Response 1: We searched all the researches including GYN-CS and actually there are three more manuscripts except our study and in the discussion part, we have mentioned about the outcomes of the studies. I would kindly show you that part in our manuscript between the lines 216 and 220. I attach the part below on your behalf. The references 19, 26 and 27 refer to previous GYN-CS studies for comparison.
"In a large scale study (n=140), with the same follow-up period as this study, 1.4% of expulsions for frameless devices and 11.4% for conventional IUDs were reported [19], while another study showed 3.6% expulsion of frameless devices as compared to 22.2% with T- shaped IUDs [26],and another study declared 1% expulsion of frameless devices[27]."